# Molecular dynamics simulations of the evaporation of hydrated ions from aqueous solution

Philip Loche [1,2], Douwe J. Bonthuis [3] & Roland R. Netz [2✉]

Although important for atmospheric processes and gas-phase catalysis, very little is known about the hydration state of ions in the vapor phase. Here we study the evaporation energetics and kinetics of a chloride ion from liquid water by molecular dynamics simulations. As chloride permeates the interface, a water finger forms and breaks at a chloride separation of ≈ 2.8 nm from the Gibbs dividing surface. For larger separations from the interface, about 7 water molecules are estimated to stay bound to chloride in saturated water vapor, as corroborated by continuum dielectrics and statistical mechanics models. This ion hydration significantly reduces the free-energy barrier for evaporation. The effective chloride diffusivity in the transition state is found to be about 6 times higher than in bulk, which reflects the highly mobile hydration dynamics as the water finger breaks. Both effects significantly increase the chloride evaporation flux from the quiescent interface of an electrolyte solution, which is predicted from reaction kinetic theory.

[1] Laboratory of Computational Science and Modeling, IMX, École Polytechnique Fédérale de Lausanne, 1015 Lausanne, Switzerland. [2] Fachbereich Physik, Freie Universität Berlin, 14195 Berlin, Germany. [3] Institute of Theoretical and Computational Physics, Graz University of Technology, 8010 Graz, Austria. ✉email: rnetz@physik.fu-berlin.de

ons in aqueous solution have a very low vapor pressure due to their favorable solvation[1–5] but nevertheless are known to be present in the earth's atmosphere, with important consequences for climate, precipitation, and atmospheric chemistry phenomena such as the regulation of the ozone concentration in the tropo- and stratospheres[6–12]. The traditional Born estimate of the ion solvation free energy in water[13] neglects ion hydration in the vapor phase, i.e., the possibility that a few water molecules adsorb from vapor onto an ion. Clearly, ion hydration in the vapor phase will substantially decrease the magnitude of the solvation free energy and therefore significantly increase the equilibrium ion concentration in the vapor phase and accelerate the ion-evaporation kinetics from aqueous electrolyte solutions.

Previous simulation studies of halide ions at the air–water interface showed that the repulsion from the interface decreases with halide size[14–17] and that ions, when they penetrate the interface from the waterside, significantly deform the interface and produce an extended water finger[18]. For water–dichloroethane and water–nitrobenzene interfaces it was shown that when a chloride ion leaves the aqueous phase, it drags a few water molecules into the organic liquid phase[19–21], which suggests that in these water-saturated organic liquids, chloride ions are present as nano-hydrated droplets. For the important case of fully or partially saturated water vapor, it is not clear whether ions are hydrated, i.e., surrounded by an adsorbed water layer, or not. Experiments indicate significant chloride content of maritime air[6,7], but without knowing the solvation free energy of hydrated chloride ions, it is impossible to estimate the relevance of equilibrium ion-evaporation versus nonequilibrium ion-evaporation mechanisms, the latter being caused by oceanic surface waves and concomitant spray and droplet formation.

For estimating the evaporation flux of ions from electrolyte solutions into the vapor phase, one needs in addition to the barrier energy, which corresponds to the ion solvation free energy, also the ion diffusivity in the transition state. This follows from the fact that it is the transition-state diffusivity that determines the reaction rate, not the bulk diffusivity. Since the ionic transition state is located close to the air–water interface, the transition-state ion diffusivity is not necessarily given by the ion bulk diffusion coefficient. By a combination of four different equilibrium and nonequilibrium simulation protocols, we determine the hydration state of an isolated chloride ion in saturated water vapor and extract both the spatial chloride solvation free energy profile across the electrolyte–vapor interface and the chloride diffusion coefficient at the transition state from simulation trajectories. This way, we gain a full understanding of the equilibrium ion hydration in the vapor phase and the evaporation kinetics from a quiescent air–water interface.

We find that a chloride ion is hydrated by about seven water molecules in close-to-saturated water vapor, which drastically decreases the ion solvation free energy by about 26 $k_BT$ from 142 $k_BT$ in dry air, which is the normally assumed reference state[22], down to 116 $k_BT$. This simulation result is corroborated by continuum-dielectric and statistical-mechanics model calculations. Further, we determine the effective chloride diffusion coefficient $D_{tr}$ in the transition state, in which the ion is connected by a thin water neck to the aqueous phase at a distance of about 2 nm from the Gibbs dividing surface (GDS). We find $D_{tr}$ to be increased by a factor of about 6 compared to the bulk value, which reflects the highly dynamic hydration structure of the water neck. Both effects, the reduction of the solvation free energy and the increase of the effective diffusion coefficient, significantly increase the chloride evaporation flux from electrolyte solutions. Still, the estimated chloride evaporation flux from the oceans is many orders of magnitude smaller than the one needed to account for the current atmospheric chloride concentration. We conclude that atmospheric chloride is released by nonequilibrium mechanisms connected to ocean surface-wave induced spray and droplet formation.

## Results and discussion

As shown previously, ions form water fingers with the water phase as they penetrate the water–vapor interface[18] or the water–organic liquid interface[19–21]. However, the possibility of ion hydration far in the vapor phase has remained unclear. We first present two simple analytical models which suggest that ions should be hydrated in saturated water vapor.

**Dielectric continuum model for ion hydration.** The first model is based on continuum dielectric theory: We assume that the ion with radius $R$ and valency $q$ (in units of the elementary charge e) is surrounded by a dielectric shell with radius $B$ and relative dielectric constant $\varepsilon$, see Fig. 1a for a schematic illustration. The free energy difference $U_\varepsilon$ between the hydrated and non-hydrated states according to the Born model on the linear dielectric response level is given by

$$\frac{U_\varepsilon}{k_BT} = \frac{q^2 l_B^{vac}}{2}\left(\frac{1}{R}-\frac{1}{B}\right)\left(\frac{1}{\varepsilon}-1\right) + 4\pi B^2 \frac{\gamma}{k_BT}, \quad (1)$$

where $l_B^{vac} = e^2/(4\pi\varepsilon_0 k_BT)$ is the Bjerrum length in vacuum and $\gamma$

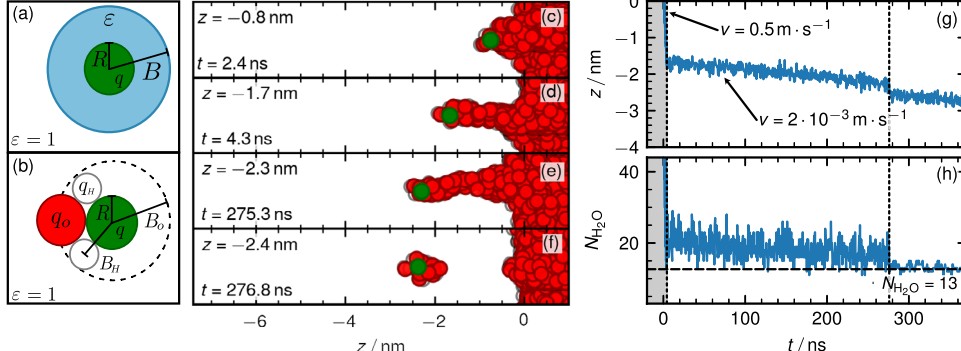

**Fig. 1 Ion pulled at a constant speed through the interface. a** Dielectric model for an ion with valency $q$ and radius $R$ embedded in a spherical droplet with dielectric constant $\varepsilon$ and radius $B$. **b** Molecular model for the binding of a water molecule to an ion. **c–f** Simulation snapshots of protocol (I) where chloride is held in a harmonic potential that moves with a prescribed velocity $v$ through the interface. **g** Chloride position and (**h**) number of water molecules $N_{H_2O}$ in a sphere with radius 1 nm around the ion as a function of time. The left vertical dotted line shows where the velocity of the potential is reduced, the right vertical dotted line shows where the water finger breaks. The horizontal broken line in (**h**) indicates the number of bound water molecules after the finger has broken. A simulation video is shown in Supplemental Movie 1 and online https://youtu.be/ygqVOVVSju8.

is the air–water surface tension. The first term denotes the dielectric polarization energy difference and the second term in Eq. (1) is the surface free energy. Note that we assume the surface tension to be independent of the radius, which is an approximation that not necessarily holds on nanoscopic length scales[23]. The optimal hydration layer radius $B^*$ follows from $dU_\varepsilon/dB = 0$ as

$$B^* = \left[\frac{q^2 k_B T l_B^{vac}}{16\pi\gamma}\left(1 - \frac{1}{\varepsilon}\right)\right]^{1/3}, \qquad (2)$$

while the free energy at the optimal value $B^*$ is given by

$$\frac{U_\varepsilon^*}{k_B T} = -\frac{q^2 l_B^{vac}}{2R}\left(1 - \frac{3}{2}\frac{R}{B^*}\right)\left(1 - \frac{1}{\varepsilon}\right). \qquad (3)$$

For the SPC/E water model applied within this study $\gamma = 54.0 \text{ mN} \cdot \text{m}^{-1}$, $\varepsilon = 70$ and $l_B^{vac} = 56$ nm obtained from our simulations in agreement with previous studies[24,25]. The optimal hydration radius for $q = \pm 1$ then is $B^* \approx 0.44$ nm. From the free energy given in Eq. (3) it follows that for ion radii smaller than the critical value $R^* = 2B^*/3 \approx 0.3$ nm the hydration shell is stable and thus water molecules should assemble around an ion. On an approximate level, the radius of a chloride ion can be estimated from the simulated or experimental solvation free energy (which perfectly agree since the experimental solvation free energy is used as an optimization target for reliable ion force fields) using the Born expression for the dielectric polarization energy similar to Eq. (1), from which one obtains $R_{Cl} = 0.19$ nm[22]. Of course, this estimate for the radius makes a number of approximations that are not really fulfilled, primarily the assumption of linear dielectric response[22], but is consistent with the linear expression Eq. (1). Since the estimated chloride ion radius $R_{Cl}$ is smaller than the critical radius $R^*$, we conclude that based on this simple dielectric continuum model, a chloride ion should be surrounded by a water shell in saturated water vapor. However, the predicted thickness of this hydration shell, $B^* - R_{Cl} \approx 0.25$ nm, is not much larger than the size of a water molecule and continuum modeling might not be accurate. Besides, the free energy expression Eq. (1) neglects the entropy loss upon water adsorption from the vapor onto the ion.

**Statistical mechanics model for ion hydration**. In an alternative molecular statistical mechanics model, we describe a single hydrating water molecule by the partial oxygen charge, $q_O$, located at a distance $B_O$ and the partial hydrogen charges $q_H = -q_O/2$ at a distance of $B_H$ from the ion center, see Fig. 1b. The hydration free energy is the sum of the Coulomb free energy and the translational and rotational entropy losses upon water binding to a chloride ion,

$$\begin{aligned}\frac{U_{bind}}{k_B T} &= \frac{U_{Coul}}{k_B T} + S_{trans} + S_{rot}\\ &= -l_B^{vac}|qq_O|\left(\frac{1}{B_H} - \frac{1}{B_O}\right) + \ln\left(\frac{v_{vap}}{v_{hyd}}\right) + S_{rot},\end{aligned} \qquad (4)$$

where $v_{vap} = 1/c_{vap} = 1415 \text{ nm}^{-3}$ is the water molecular volume in fully saturated vapor and $v_{hyd} = 4\pi B_O^3/3$ is the approximate hydration volume, i.e., the volume available for a hydrating water molecule. Estimating the chloride-oxygen distance as $B_O = 0.322$ nm from the chloride-water radial distribution function (see Supplementary Fig. 1) and $B_H = 0.27$ nm using the SPC/E geometric parameters, we find from the SPC/E partial charges[26] $U_{Coul}/k_B T \approx -24$. The translational entropy loss is $S_{trans} \approx 10.83$, while the rotational entropy loss upon water binding has been estimated as $S_{rot} \approx 2$ for a similar system[27]. In conclusion, the Coulomb free energy in Eq. (4) outweighs the entropy loss and we estimate a favorable hydration free energy of about $U_{bind} \approx -11$

$k_B T$ for the first water molecule that adsorbs onto a chloride ion (note that the binding free energy of subsequently adsorbing water molecules will of course be reduced due to Coulomb repulsion). This corroborates the conclusion from the dielectric continuum model that chloride should be hydrated in saturated water vapor. Clearly, for a quantitative analysis, simulations are needed.

**Ion evaporation from molecular dynamics simulations**. All details of the molecular dynamics simulations are explained in "Methods". Figure 1c–f shows simulation snapshots using protocol (I), where the chloride ion is confined in a harmonic potential that moves at prescribed velocity $v$ from the liquid to the vapor phase (the vapor phase is located at $z < 0$ and the water phase at $z > 0$, where $z$ denotes the position relative to the GDS). For $z > -1.7$ nm, in which range a stable water finger is present, we move the potential at a relatively fast velocity of $v = 0.5 \text{ m} \cdot \text{s}^{-1}$ (gray area in Fig. 1g, h). For $z < -1.7$ nm, where the water finger is becoming unstable, we reduce the velocity by a factor of 250 down to $v = 2 \cdot 10^{-3} \text{ m} \cdot \text{s}^{-1}$. Figure 1g shows the $z$ position of the confined ion and Fig. 1h the number of water molecules $N_{H_2O}$ inside a sphere of radius 1 nm around the ion as a function of time. At $z \approx -2.4$ nm the water finger breaks, and the ion position jumps away from the interface, which demonstrates that the water finger pulls on the ion. After the finger breaks, $N_{H_2O} = 13$ water molecules stay bound to the ion. The position of the breakdown and the number of bound water molecules depend on the velocity with which the potential moves (results for different velocities are shown in Supplementary Fig. 2).

To study the long-time dynamics of the ion hydration and the water–finger break up, we use simulation protocol (II), where the ion-confining potential is fixed in space. Figure 2a–f shows snapshots for two potential positions $z = -2.8$ nm and $z = -7.0$ nm starting from an initial state with no water molecules adsorbed to the ion. For $z = -2.8$ nm the ion attracts water molecules from the nearby interface, and a water finger forms and breaks periodically. For $z = -7.0$ nm the ion forms a hydration shell as well but this time no water finger is present, the water molecules are in fact captured from the vapor phase. The number of hydration waters $N_{H_2O}$ as a function of time in Fig. 2g demonstrates the oscillatory water–finger dynamics with a period of about 150 ns for $z = -2.8$ nm and the formation of the ion hydration shell over a relaxation time of about 50 ns for $z = -7.0$ nm.

We fit the distribution of the number of water molecules $N_{H_2O}$ in the ionic hydration shell for $z = -7.0$ nm in Fig. 2h by a Gaussian

$$P(N_{H_2O}) = \frac{1}{\sigma\sqrt{2\pi}}e^{-\frac{1}{2}\left(\frac{x-\mu}{\sigma}\right)^2}, \qquad (5)$$

where $\mu = 6.59 \pm 0.03$ nm and $\sigma = 0.93 \pm 0.03$ nm are the mean and the standard deviation. The non-hydrated state is never observed in the simulations, except in the initial equilibration stage. We, therefore, estimate the free energy of the non-hydrated ion state from the Gaussian fit by extrapolation as $U_{hyd}/k_B T \approx \ln P(0) = -26 \pm 3$. With the limited simulation data for the hydration number distribution, the accuracy of this Gaussian approximation is difficult to assess, but we note that the result is not inconsistent with the free energy $U_{bind} \approx -11 \text{ } k_B T$ from Eq. (4) for the adsorption of a single water molecule to a chloride ion, keeping in mind that the subsequent adsorption of water molecules will be less favorable due to shielding of the ionic charge.

Knowing the ion hydration structure at the interface and in the vapor phase from simulation protocols (I) and (II), we now

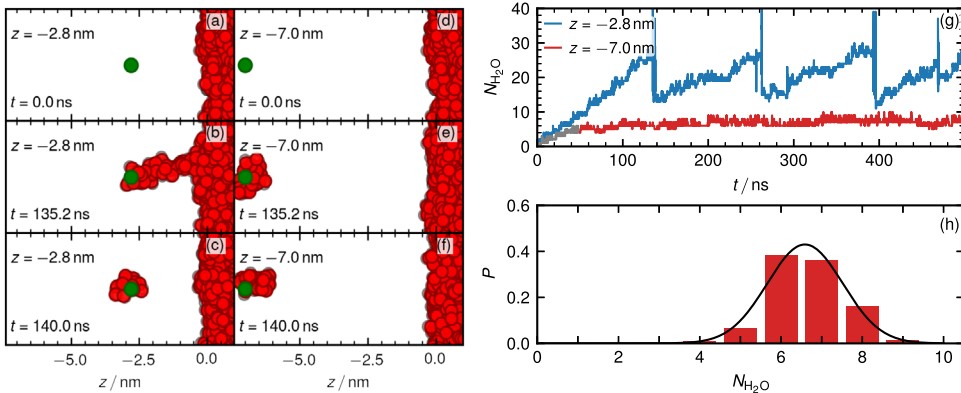

**Fig. 2 Ion held at fixed position.** Simulation protocol (II) where an ion is confined to a harmonic potential fixed at different $z$ positions. Simulation snapshots at different times are shown for $z = -2.8$ nm (**a–c**) and $z = -7.0$ nm (**d–f**). **g** Number of water molecules $N_{H_2O}$ in a sphere with radius 1 nm around the ion for two different positions as a function of time, initially $N_{H_2O} = 0$. **h** Distribution of $N_{H_2O}$ for $z = -7.0$ nm, where the first 50 ns of the simulation are discarded for equilibration. The black solid line is a Gaussian fit from which the free energy for $N_{H_2O} = 0$ is obtained by extrapolation.

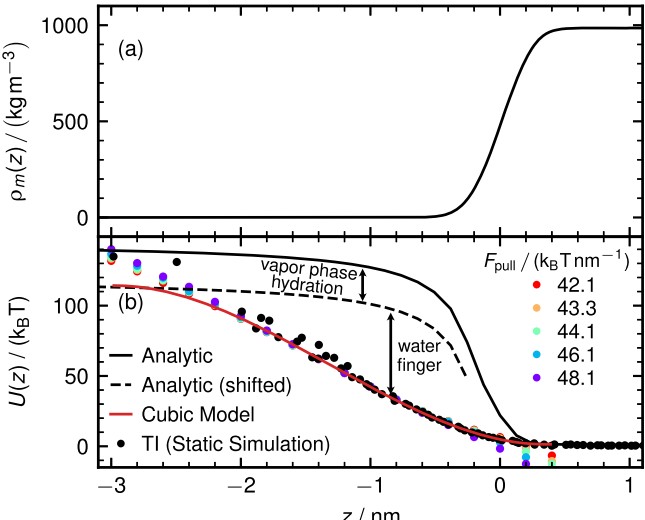

**Fig. 3 Ion free energy from thermodynamic integration (TI). a** Water mass density profile $\rho_m(z)$ of the water–vapor interface. **b** Chloride free energy profile $U(z)$ obtained by TI (protocol (III), black spheres) where the ion is held at different fixed positions. The error bars are of the same size as the data points. The black solid line shows the analytic image-charge potential for a charged sphere at a sharp flat dielectric interface[29], the black dashed line is shifted by the vapor hydration free energy $U_{hyd} = -26\,k_BT$. The red solid line is a cubic polynomial fit to the TI data in the range $-2.0$ nm $< z < 0.4$ nm. Colored dots show the free energy obtained from the logarithm of the ion distribution of protocol (IV) simulations at constant force $F_{pull}$.

determine the position-dependent hydration free energy from TI using the protocol (III). Figure 3a shows the water mass density profile $\rho_m(z)$ across the air–water interface, which exhibits the typical shape for a vapor-liquid interface and is broadened due to capillary surface waves[28]. Figure 3b shows the free energy profile $U(z)$ for a $Cl^-$ ion. The TI data is shown as black spheres, where in the water slab center ($z = 5$ nm) we set $U(z) = 0$. We subtract from our simulation results the interaction between the chloride ion and the counter ion in the water slab and all periodic images using analytic expressions[22,29], as shown in Supplementary Section 3. Colored spheres show results from simulations at constant pull force $F_{pull}$ (simulation protocol IV, explained in more detail below), obtained from the negative logarithm of the distribution function and shifted such as to coincide with the TI

data at $z = -1$ nm, which exhibits good agreement with the TI data. In Supplementary Fig. 6, we present umbrella-sampling results for $U(z)$, which are based on shorter simulations than our TI data, and as a consequence exhibit significant deviations from our TI results. Since our TI results agree rather accurately with constant pull force simulations from protocol (IV), as shown in Fig. 3b, we trust our TI data and therefore did not use umbrella-sampling results in our analysis. The black solid line denotes the analytical expression for the image–charge interaction of a charged sphere of radius 0.19 nm with a sharp dielectric interface at $z = 0$ nm, which saturates to a constant value at separations $z < -10$ nm, as shown in Supplementary Fig. 4a, see Supplementary Section 3 for a derivation of the analytical expression[29]. In the derivation of the analytical expression, we use only one dielectric interface, whereas in our simulations we have a slab system with two interfaces. However, as we show in Supplemental Fig. 5, we find that for a water slab thickness of 10 nm, as in our simulations, the image–charge interaction for a slab system agrees very well with the result for a single interface. The TI free energy agrees with the analytic expression (black solid line) for large distances from the surface of about $z \approx -3$ nm. This is not surprising, as the analytical image–charge theory neglects the interface deformation, the water finger formation, and ion hydration in the vapor phase. On the other hand, the TI, which always starts from an unhydrated ion and consists of simulations with a duration between 5 and 100 ns per integration step, does for large separations $z < -2.0$ nm from the interface not yield the correctly hydrated ion state, which has a somewhat longer equilibration time, as shown in Fig. 2g. As a consequence, the TI in Fig. 3b is expected to follow the image–charge theory for an unhydrated ion (solid black line) for $z < -3$ nm in a smooth fashion as long as the TI simulations trajectories are not longer than the time needed for the hydration shell to form from the vapor phase, which is about 100 ns. The TI data can therefore only be trusted for $z \geq -2.0$ nm, where the time needed to establish the water finger is safely reached within the TI simulation time. The image–charge potential shifted by the ion hydration free energy $U_{hyd} = -26\,k_BT$, denoted by a black dashed line, should describe the ion free energy accurately when the water finger is absent and therefore serves as an upper bound for the true ion free energy profile $U(z)$.

To obtain an analytic form of $U(z)$, we fit a cubic polynomial to the TI data in the range $-2.0$ nm $< z < 0.4$ nm, where the TI data can be trusted. This fit, shown as a red solid line, is constructed such as to reach the broken black line at $z = -2.8$ nm, the distance at which the water finger is seen to become unstable

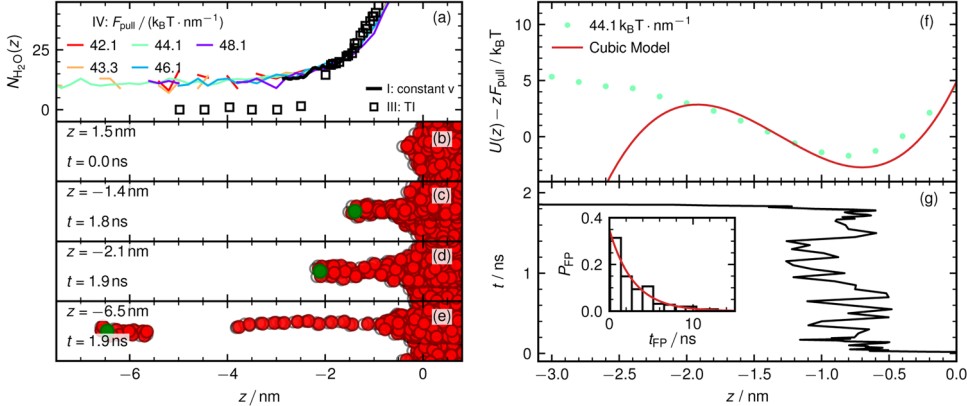

**Fig. 4 Ion dynamics with externally applied force.** Simulation protocol (IV) where a constant force $F_{pull}$ acts on the chloride ion. **a** Number of water molecules $N_{H_2O}$ in a spherical sphere with radius 1 nm around chloride as function of position. Colored lines are simulations at constant force $F_{pull}$. The black line shows results from constant velocity simulations (protocol I, Fig. 1h) and black open squares show TI simulations (protocol III, Fig. 3b). **b**–**e** Snapshots at different ion positions for $F_{pull} = 44.1\,k_B T \cdot nm^{-1}$. **f** Free energy for $F_{pull} = 44.1\,k_B T \cdot nm^{-1}$, green dots show results from the ion distribution of 200 simulation runs, the red solid line is the cubic polynomial fit from Fig. 3b. **g** Ion trajectory corresponding to snapshots in (**b**–**e**). The inset shows a histogram of first passage times for 200 simulation runs with an exponential fit (solid red line). A simulation video of an evaporating chloride ion is available in Supplemental Movie 2 and online https://youtu.be/LVAHxClnzlU.

from the oscillatory behavior in Fig. 2g, and to reproduce the bulk free energy $U(z) = 0$ at $z = 0.4$ nm with vanishing slope. In conclusion, the deviation of the true ion free energy from the analytical image–charge prediction (black solid line) for $z < -2.8$ nm is caused by ion hydration and for $z > -2.8$ nm by water finger formation. The fitted red line accounts for both effects.

To characterize the ion-evaporation kinetics, we apply a constant force $F_{pull}$ on the ion (protocol IV). Figure 4a shows $N_{H_2O}$ as a function of position for different $F_{pull}$ (colored lines) in comparison with results from TI (protocol III, black squares) and constant velocity simulations (protocol I, black line). The results agree closely except for the TI results for $z < -2$ nm, for reasons that were discussed earlier. Figure 4b–e shows simulation snapshots at different times for a force $F_{pull} = 44.1\,k_B T \cdot nm^{-1}$, in agreement with our constant position simulations in the protocol (II) shown above, the ion drags a few water molecules along as it evaporates.

Figure 4f shows the free energy profile $U(z) - zF_{pull}$ from the cubic fit for $U(z)$ in Fig. 3b for a force $F_{pull} = 44.1\,k_B T \cdot nm^{-1}$ (solid line). The green dots show the free energy obtained from the logarithm of the ion position distribution from simulations at constant force averaged over 200 simulation runs, which agrees with the red line quite well to the right of the barrier, which demonstrates the consistency of the different simulation protocols (the disagreement to the left of the barrier reflects the nonequilibrium character of the trajectories which leave the water phase and never return from the vapor phase). The applied force reduces the energy barrier height $\Delta U$ from $\approx 110\,k_B T$ to $\approx 4.5\,k_B T$, so that evaporation events are frequently observed in the simulations. In Fig. 4g, we present an ion-evaporation trajectory for $F_{pull} = 44.1\,k_B T \cdot nm^{-1}$. Initially, the ion is in the center of the water slab at $z = 5$ nm. Due to the applied force, it quickly moves towards the free energy minimum around $z = -0.8$ nm and crosses the barrier after a first passage or waiting time of about $t_{FP} = 1.8$ ns. The inset shows the normalized first-passage time distribution $P_{FP}$ for 200 evaporation events. The red solid line is an exponential fit according to $P_{FP}(t_{FP}) = e^{-t_{FP}/\tau_{MFP}}/\tau_{MFP}$, where the mean first-passage time is $\tau_{MFP} = 2.9$ ns. We note that we chose a rather large lateral size of the simulation box of $5.1\,nm \times 5.4\,nm$, but we cannot exclude that finite-size effects influence the kinetics of the observed ion-evaporation events, so our results for $\tau_{MFP}$ could contain a systematic error.

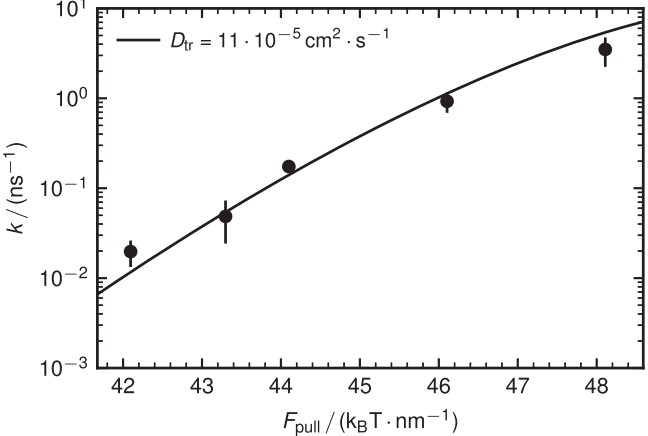

**Fig. 5 Ion-evaporation rate.** Evaporation rate $k$ of a chloride ion from protocol (IV) simulations as a function of the applied force $F_{pull}$, error bars denote the standard deviation of independent simulations runs divided by the square root of their number. The solid line shows the Kramers rate Eq. (7) in the limit $L \to 0$ with a fitted transition-state diffusion constant $D_{tr} = 11 \cdot 10^{-5}\,cm^2 \cdot s^{-1}$.

**Reaction rate theory for ion-evaporation kinetics.** In Fig. 5, we show the simulated evaporation rate $k = 1/(2\tau_{MFP})$ as a function of the applied force $F_{pull}$ (data points). The reaction rate theory used to describe the ion-evaporation kinetics is explained and derived in the Methods section and in Supplementary Section 5. The solid black line corresponds to the Kramers rate Eq. (7) in the limit $L = 0$, since in the simulations with applied force $F_{pull}$ the potential well is characterized by the curvature $U''_{min}$ and no additional reservoir of length $L$ is present. The values for the potential barrier $\Delta U$ and the curvatures $U''_{min}$ and $U''_{max}$ follow from the cubic fit function shown in Fig. 3b. In Supplementary Section 6, we show that the ion barrier crossing kinetics is dominated by the ion diffusivity at the transition state and not by the bulk ion diffusivity. The single-parameter fit yields the ion diffusion coefficient at the transition state $D_{tr} = 11 \cdot 10^{-5}\,cm^2 \cdot s^{-1}$, which is 6 times larger than the chloride bulk diffusivity $D_{bulk} = 2 \cdot 10^{-5}\,cm^2 \cdot s^{-1}$, which is determined from independent simulations and which agrees well

with the experimental result $D_{bulk,exp} = 2.24 \cdot 10^{-5}\,cm^2 \cdot s^{-1}$[30]. In the transition state, at the barrier top around $z \approx -2\,nm$, the ion is only loosely hydrated within the water finger, see Fig. 4b–e, which explains the significantly increased diffusivity compared to bulk.

Knowing the free energy profile $U(z)$ and the ion diffusivity at the transition state $D_{tr}$, we can calculate the chloride evaporation flux $J$ from the earth's oceans using Eq. (8). So far our simulations and calculations were for a single ion. A finite ion concentration as present in the ocean will screen the Born solvation energy and the image–charge interaction within the liquid phase[31]. The effect on the solvation free energy can be described by the concentration-dependent activity coefficient $\gamma$, which for a NaCl solution at $c_{ocean} = 599\,mol \cdot m^{-3}$, the salt concentration of the oceans, takes a value of $\gamma = 0.72$[32]. From this, the salt concentration-dependent correction to the ion solvation free energy amounts to $\ln\gamma = -0.32$ in units of $k_B T$, which is negligible compared to the much larger single-ion solvation free energy. From the area of the earth's oceans $A_{ocean} = 3.61 \cdot 10^{14}\,m^2$, we estimate an ion flux of

$$\begin{aligned} J &= A_{ocean}\,c_{ocean}\,k_0\,e^{-\frac{\Delta U}{k_B T}} \\ &= 2.22 \cdot 10^{-23}\,mol \cdot y^{-1}, \end{aligned} \qquad (6)$$

where we used $k_0 = 33\,m \cdot s^{-1}$ and $\Delta U = 113\,k_B T$ as derived in Supplementary Section 5. Neglecting the return to the oceans and the loss to higher altitudes, the current chloride concentration in the earth's atmosphere due to evaporation is predicted to be $\approx 10^{-32}\,mol \cdot m^{-3}$, where we used the age of the earth as $4.54 \cdot 10^9$ years and the height of the troposphere as $\approx 12\,km$[33]. This predicted concentration is many orders of magnitude smaller compared to the actual present concentration of chloride in the atmosphere, which is $1.3 \cdot 10^{-7}\,mol \cdot m^{-3}$ (see ref. [7]). We conclude that chloride evaporation from the quiescent air-ocean interface cannot explain the present chloride concentration in the atmosphere, which rather must be caused by spray formation due to oceanic surface waves and wind activity[9,10,12,34–36].

**Conclusion.** Using a combination of four different molecular dynamics simulation methods, we investigate the evaporation of chloride ions from the air–water interface. The simulations reveal that in saturated water vapor, chloride ions take seven water molecules with them as they evaporate from the water phase, as follows from long simulations of a chloride ion in the vapor phase and is checked by simple dielectric-continuum and statistical mechanics models. Clearly, the number of hydration water will decrease with decreasing relative humidity. This hydration shell significantly lowers the magnitude of the solvation free energy of chloride ions and thus increases the equilibrium concentration of chloride ions in the vapor phase. The effect of the ion hydration is typically excluded in estimates of ionic solvation free energies since usually a cycle is employed where the solvation free energy of neutral compounds is used.

We also determine the chloride diffusivity in the transition state $D_{tr}$, which is increased by a factor of 6 compared to the bulk value due to the formation of a highly dynamic water finger that engulfs the ion. The reduction of the solvation free energy together with the increased transition state diffusivity increases the spontaneous evaporation kinetics of chloride ions from electrolyte solutions significantly, though not enough to explain the current atmosphere chloride concentration, which must be caused by spray and droplet formation induced by oceanic waves and wind activity. Nevertheless, the spontaneous chloride evaporation across air–water interfaces is important for understanding the balance of hydrated ions and electrolyte droplets in the earth's atmosphere. In particular, our prediction that chloride ions are surrounded by a hydration shell in saturated water vapor

is important for atmospheric chemical reactions including rain and cloud formation and awaits experimental confirmation.

Similar effects as we find for chloride are also expected for other ions and in particular for multivalent ions. Previous studies investigated the evaporation mechanism of water molecules from the vapor-liquid water interface[37]. Compared to ions, water molecules evaporate as single molecules and do not form extended water fingers. Also, the free-energy barrier of water evaporation is just a few $k_B T$, in accordance with the much higher vapor pressure of water. As a consequence, water evaporation from an interface is a frequent process that can be easily observed in simulations, in contrast to the evaporation of ions.

The importance of atmospheric chloride and its hydration state originates from the fact that it regulates the ozone concentration[38,39], which in turn is crucial for sustaining life on earth. Most of the ozone is located in the stratosphere ($\approx$12–55 km), where it absorbs short-wave radiation from the sun and thus protects living organisms from potentially fatal genetic damage[40]. Only about 10% of the total ozone is located in the troposphere (below $\approx$12 km) where ozone is the source of hydroxyl (OH) radicals, which is the most important oxidizing agent responsible for the self-cleaning capacity of the atmosphere[8]. The ozone concentration also influences antarctic temperature changes and the southern hemisphere mid-latitude circulation[41–43]. Therefore, the experimental and theoretical study of atmospheric halides and their hydration state is directly relevant for the ozone balance and indirectly to climate change and global warming phenomena.

## Methods

**Evaporation kinetics model.** For an overdamped system governed by a potential $U(z)$ along a one-dimensional reaction coordinate, the rate of reaching a barrier from a reservoir of length $L$ as derived in Supplementary Section 5 is given by

$$k = \frac{1}{2\tau_{MFP}} = \frac{k_0}{\sqrt{\frac{2\pi k_B T}{U''_{min}}} + L}\,e^{-\frac{\Delta U}{k_B T}}, \qquad (7)$$

where $\tau_{MFP}$ is the mean first passage time of reaching the barrier top, $U''_{min}$ is the potential curvature at the potential minimum and $\Delta U = U_{max} - U_{min}$ is the barrier height, i.e., the difference between the potential maxima and minima. The reaction rate coefficient is given by $k_0 = D_{tr}\sqrt{U''_{max}/(2\pi k_B T)}$ and has units of a velocity, where $D_{tr}$ is the diffusion coefficient in the transition state, i.e., the barrier top (see Supplementary Section 6). For $L \to 0$ in Eq. (7) we recover the Kramers result in the overdamped limit[44], while for $L \to \infty$, the relevant limit for evaporation from a bulk reservoir, the barrier crossing rate is given by

$$\lim_{L\to\infty} k = \frac{k_0}{L}\,e^{-\frac{\Delta U}{k_B T}} = \frac{D_{tr}}{L}\sqrt{\frac{U''_{max}}{2\pi k_B T}}\,e^{-\frac{\Delta U}{k_B T}}. \qquad (8)$$

**Simulation model.** Classical non-polarizable force-field molecular dynamics simulations are performed using the GROMACS package[45]. For water, we use the SPC/E model. The chloride parameters are taken from ref. [46], which are close to recently optimized parameters that well reproduce experimental solvation free energies, activity coefficients, dielectric constants, mass densities, and conductivities over a large concentration range[47]. The used chloride force field has also been shown to reproduce the experimental ion excess at the air–electrolyte interface[48], which suggests that it is not necessary to use polarizable or charge-scaled force fields[49] for the present study. The periodic simulation box has an extension of $5.1 \times 5.4 \times 70$ nm and contains 9024 water molecules at a temperature of $T = 300$ K that spontaneously form a slab with a thickness of roughly 10 nm thickness and a vapor phase with a thickness of roughly 60 nm. The vapor phase is fully saturated by construction and has a water number density of $c_{vap} = 7 \cdot 10^{-4}\,nm^{-3}$, obtained from simulations without an ion, which corresponds to an ideal gas pressure of $P_{vap} = 30$ mbar, slightly lower than the experimental water vapor pressure of $P_{vap} = 42$ mbar at 300 K[33]. The system is periodically replicated in all directions. Besides the water molecules, we add a chloride ion with a charge $-e$ and a counter ion with a charge $e$ that is fixed in the center of the water slab, which avoids artifacts in charged inhomogeneous systems[50]. The counter ion corresponds to a lithium cation and has a substantially larger solvation free energy than chloride, which prevents the formation of a second water slab around the chloride ion. As a consequence, the water vapor humidity is slightly decreased from full saturation. To furthermore avoid motion of the water slab, we apply a harmonic force on the

center of mass of the system in the $z$ direction with a force constant $1000\,\mathrm{kJ} \cdot \mathrm{mol}^{-1} \cdot \mathrm{nm}^{-2}$. We employ four different simulation protocols. In protocol (I) we confine the chloride ion inside a harmonic potential which is moving with a constant velocity along the $z$ direction. In protocol (II) the chloride ion is confined inside a harmonic potential that is fixed in space. In protocol (III) we determine the free energy profile using thermodynamic integration (TI), where the chloride ion is frozen at different positions. Here, the simulation length per integration step is between 5 and 100 ns. In Supplementary Fig. 6, for comparison, we also show results from umbrella-potential simulations with much shorter trajectories. In protocol (IV) we apply a constant force $F_{pull}$ on the chloride ion perpendicularly to the air–water interface. More detailed information about all simulation methods is given in Supplementary Section 1.

## Data availability

The datasets generated and analyzed during the current study are available from the corresponding author on reasonable request. Supplementary Movie 1 shows a chloride ion pulled from an air water interface with a constant velocity of $300\,\mathrm{m} \cdot \mathrm{s}^{-1}$. Supplementary movie 2 shows the evaporation process of a chloride ion where a constant force of $44\,\mathrm{k_B T}$ pointing from the water to air phase is acting on the ion.

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

## Acknowledgements

We gratefully acknowledge support by the MaxWater initiative from the Max-Planck Society; the Deutsche Forschungsgemeinschaft (DFG) via Grant IRTG-2662 "Charging into the future: Understanding the interaction of polyelectrolytes with biosystems" contract number 434130070; computing time on the HPC cluster at ZEDAT, FU Berlin.

## Author contributions

P.L. performed and analyzed the simulations. P.L. and R.R.N. performed the theoretical calculations. P.L. designed the figures and movies with input from all authors. P.L., D.J.B., and R.R.N. wrote the paper, provided critical feedback, and participated in scientific discussions.

## Funding

## Competing interests

The authors declare no competing interests.
