## [Peer Review File · Communications Chemistry]

Reviewers' comments:

Reviewer #1 (Remarks to the Author):

In this manuscript, the authors estimate the rate at which a chloride ion evaporates from liquid water into saturated water vapor using molecular simulations and enhanced sampling calculations.

The authors find that chloride ion does not lose all its hydration waters upon evaporation but retains roughly 7 waters in the vapor phase. Consistent with previous studies, the authors show that ion evaporation is mediated by the formation of a water finger, and further show that the water finger becomes unstable roughly 2.8 nm away from the water-vapor interface.

To predict the overall rate of ion evaporation, the authors estimate the free energetic barrier for evaporation as well as the effective diffusivity of the chloride ion in the transition state. They find that the latter is roughly 6 times the diffusivity of the ion in bulk water.

The authors further find that the estimated kinetics of chloride evaporation (from a quiescent interface) are too slow to rationalize the chloride evaporation flux from the earth's oceans. They speculate that spray formation due to wind and oceanic waves must be the dominant contributors to atmospheric chloride concentration.

Overall, this manuscript tackles an interesting and challenging problem of ion evaporation from aqueous solutions and is likely to be of interest to the broad Communications Chemistry audience as well as physical and theoretical chemistry experts on ion hydration. The manuscript is also well written and easy to understand. I am thus happy to recommend its publication once the authors have the opportunity to address the comments listed below.

1) The noise in the TI results shown in Fig. 3b for z below -1 nm (and the corresponding deviation from the smooth cubic fit) suggests the presence of substantial error in $U(z)$, and even more so in the corresponding forces that are estimated in the simulations. Given that these errors influence the estimated ion evaporation barrier, the authors are encouraged to include error bars.

Assuming that there is overlap between the umbrella sampling windows (for the free energy profiles reported in Fig. S8b), I am inclined to trust the authors' umbrella sampling results more than the TI results.

The authors should be commended for performing two sets of umbrella sampling simulations starting from different initial states, i.e., the hydrated ion in water (forward, solid green line) and the bare ion in vapor (reverse, dashed green line). The authors show that free energy profiles obtained from the two sets of simulations do not agree with one another due to the hysteresis associated with ion hydration and water finger formation; these results highlight the challenges in estimating $U(z)$.

The true $U(z)$ is likely to lie in between the forward and reverse umbrella sampling $U(z)$ profiles, i.e., the solid and broken green lines in Fig. S8, suggesting that the reversible work of evaporating the Cl is substantially lower than that suggested by the cubic fit to the TI free energy profile.

How sensitive are overall conclusions to the choice of $U(z)$, e.g., what would the predicted

atmospheric chloride concentration be if forward (and reverse) umbrella sampling results were used to estimate it instead?

Finally, the authors may also be able to reduce hysteresis in the umbrella sampling simulations by using a hydrated ion (with 7 waters rather than a bare ion) to estimate the reverse $U(z)$ profiles.

2) I could be mistaken, but a back-of-the-envelope calculation suggests that for the estimated $U(z)$, the value of $\sqrt{2 \pi kT / U''_{\min}}$ is much smaller than the simulation box size. The authors are encouraged to justify their use of eq. 1 with $L=0$ to obtain the transition state diffusivity (in Fig. 5).

3) As the authors (and others) have shown, the water finger plays a central role in the ion evaporation process. Should we expect a coupling between the distortion of the water-vapor interface in the vicinity of the water finger and the capillary wave spectrum of the interface far from the water finger? If so, might the free energy profiles determined by the authors (and therefore, the barrier to ion evaporation) depend on the cross-section of the simulation box?

4) For the water number distribution shown in Fig. 2h, do the mean and the standard deviation become independent of z below -7 nm? The authors estimate the free energetics of ion hydration assuming that this distribution is Gaussian. The authors are encouraged to explain why this is a reasonable assumption. A plot of $-\log P$ (perhaps as inset or in SI) may also be helpful in highlighting that the fluctuations are Gaussian not just near the mean, but also in the tails.

Minor Points / Typos:

1) Page 4: Fig. 2 g, h  Fig. 1 g, h.

2) Caption of Fig. 4: Two instances of $F_{\text{pull}} = 44.3$  44.1 to be consistent with text.

3) The cubic fits (red line) in Fig. 3b and Fig. S8b appear to be different.

Reviewer #2 (Remarks to the Author):

“Evaporation of Nano-Hydrated Ions from Aqueous Solution” by Netz and co-workers studies the transfer of a chloride ion from the aqueous phase to an adjacent vapor phase. The phenomenological finding of this study, that chloride ions do not spontaneously ‘evaporate’ at any appreciable rate and therefore do not contribute significantly to atmospheric chloride, seems rather obvious. A preliminary literature survey of the phase transfer thermodynamics and mechanism of chloride should reveal this outcome, e.g. Reference 17,18, and other works not included in the main text. Comparisons to related work in the academic literature are also lacking. For these reasons, the motivations and background of this work as presented by the authors are not strong. Revisiting atmospheric Cl at the end of the manuscript with related flux calculations seems forced and whimsical.

However, these simulations are interesting as an academic study of this rare event and may serve as an interesting companion piece to earlier work in the field. The simulations and analyses appear sound but again would benefit from comparison to existing work. The novelty of the results of these calculations is very low and will not significantly influence thinking in the field. Regarding potential impact, I would recommend publication in a more specialized journal.

I believe that addressing the following concerns would strengthen this work, whether resubmission

to this journal is permitted or if submitted elsewhere:

- 1) The construction of the introduction and main findings should be reconsidered, particularly considering comments above. A purely academic investigation of Cl water/vapor transfer seems far more compelling than suggesting that it may be a significant contributor to atmospheric Cl.
- 2) The 'water finger' accompanying [Cl⁻] ion transfer phenomena was described nearly 30 years ago and should be cited, e.g. *Science* 1993, 261 (5128), 1558–1560. The authors showcase this mechanism in several figures and the main text quite often. Also regarding mechanism, the authors should compare this event to the evaporation of water, which was shown to occur one water molecule at-a-time: *Phys. Rev. Lett.* 2015, 115 (23), 236102.
- 3) The Methods section in the main text contains very little information regarding the simulations. The basic class of the simulations, fixed-charge classical MD, should be clear to the reader. In its current form the simulation approach is not clear, with most information relegated to the SI. This could be remedied with minimal impact on word count by simply mentioning the water and Cl-models in the main text.
- 4) Related to (3), the authors do not consider AIMD or polarizable models, which would probably affect the results dramatically. In the current formalism, charge scaling could also be considered to compensate for the lack of polarizability, e.g. *J. Phys. Chem. Lett.* 2019, 7531–7536 and may be an interesting addition to this work.
- 5) The spontaneous breaking and reformation of the "water finger" in liquid/liquid Cl⁻ transfer has been previously reported, *J. Chem. Phys.* 2016, 145 (1), 014701, differences in water finger lengths and dynamics in liquid/liquid versus liquid/vapor may provide more mechanistic insight.
- 6) The results shown in 3b are interesting and show different approaches (TI & a simple image charge) arrive at the same state function at the beginning and end with mechanistic deviation seen in the free energy profiles. However, the corresponding discussion in the text is difficult to parse and should be revised. Also, why is the TI free energy profile still increasing at $z = -3$ nm? Is there an endpoint plateau?
- 7) Animations should be included as SI (not youtube links.)

Reviewer #3 (Remarks to the Author):

Report on "Evaporation of nano-Hydrated Ions from Aqueous Solution" by P. Loche et al.

In this work, the authors have presented theoretical studies of evaporation of a chloride ion from water surface by carrying out four different equilibrium and non-equilibrium molecular dynamics simulations. The work reveals the important result that the chloride ion evaporates in nano-hydrated form which substantially reduces the free energy barrier for evaporation and accelerates the rate of evaporation. In spite of this favorable route for evaporation of chloride ions in nano-hydrated form, the predicted concentration of chloride ions in atmosphere is still much lower than what is experimentally reported to be present over ocean. Thus, the present study also suggests that major part of chloride ions in the atmosphere come not through normal evaporation, but through non-equilibrium processes caused by oceanic waves and wind. The current results are important and, in my opinion, merits publication in *Communications Chemistry*. The following point, however, may be noted.

The current results are for SPCE model of water and charged Lennard-Jones model of ions. Thus,

none of the solvent and ions have polarizability. Earlier studies have shown that polarizability can be important for surface behavior of anions. This point may be discussed.

What could be the effects of finite concentration? For example, NaCl concentration in ocean water is about 0.6 M. It would be good to have a discussion on this concentration issue as well.

Otherwise, this is an important piece of work. The paper is written well and I support its publication.

Prof. Roland Netz

Phone: +49 30 838 55 737

Fax: +49 30 838 53741

Mail: metz@physik.fu-berlin.de

Web: <https://www.physik.fu-berlin.de/einrichtungen/ag/ag-netz>

Berlin, February 6th, 2022

Thank you very much for the reports. We thank the three reviewers for their constructive criticism and the support of our manuscript. We highlight all changes in the manuscript in red, except minor modifications. In the following, we reproduce the referee reports in full and reply to the referees' suggestions point by point.

Reviewer #1

In this manuscript, the authors estimate the rate at which a chloride ion evaporates from liquid water into saturated water vapor using molecular simulations and enhanced sampling calculations.

The authors find that chloride ion does not lose all its hydration waters upon evaporation but retains roughly 7 waters in the vapor phase. Consistent with previous studies, the authors show that ion evaporation is mediated by the formation of a water finger, and further show that the water finger becomes unstable roughly 2.8 nm away from the water-vapor interface.

To predict the overall rate of ion evaporation, the authors estimate the free energetic barrier for evaporation as well as the effective diffusivity of the chloride ion in the transition state. They find that the latter is roughly 6 times the diffusivity of the ion in bulk water.

The authors further find that the estimated kinetics of chloride evaporation (from a quiescent interface) are too slow to rationalize the chloride evaporation flux from the earth's oceans. They speculate that spray formation due to wind and oceanic waves must be the dominant contributors to atmospheric chloride concentration.

1) The noise in the TI results shown in Fig. 3b for z below -1 nm (and the corresponding deviation from the smooth cubic fit) suggests the presence of substantial error in $U(z)$, and even more so in the corresponding forces that are estimated in the simulations. Given that these errors influence the estimated ion evaporation barrier, the authors are encouraged to include error bars.

Assuming that there is overlap between the umbrella sampling windows (for the free energy profiles reported in Fig. S8b), I am inclined to trust the authors' umbrella sampling results more than the TI results.

We thank the reviewer for pointing out that the discussion of errors was incomplete in the previous version of the manuscript. Since this discussion is important to understand why we prefer our TI results over our umbrella sampling results, we added details on our error estimates of both

simulation results in the SI and also mention the error of the TI method briefly in the main text. In short: The estimated error of the TI results, calculated via error propagation from the individual errors of each integration step, is about $\sim 1 k_B T$ and thus substantially smaller than the mean deviations from the fit function in Fig. 3b, which is about $\sim 5 k_B T$ in the range $-2.0 \text{ nm} < z < -1.0 \text{ nm}$. We mention that the estimated error is smaller than the symbol size in the revised version in the caption of Fig. 3b and therefore error bars would not be visible in Fig. 3b. The difference between the calculated errors and the visible deviations from the fit function most likely comes from the slow dynamics of the water-finger fluctuations, which occurs over more than 100 ns, as shown in Fig. 2g. The estimated error of the data obtained from umbrella sampling is about $\sim 5 k_B T$ and is in the revised version shown in Fig. S8b in the Supplement, where we now present the individual data points from the WHAM analysis and not a line through the data, as done in the previous version. We used umbrella potentials with a spacing of 0.2 nm that sufficiently overlap (note that the data points in Fig. S8b result from postprocessing via the WHAM analysis and have a much smaller separation). However, when the water finger is present (for $z < -1.5 \text{ nm}$), the sampling time of 5 ns per potential in the umbrella sampling protocol is rather short. We therefore trust the TI data more, where we spent 50 ns per integration step in the distance range where the water finger is present, which results for 20 integration steps in a total of 1 μs simulation time per data point. Most importantly, the TI data is more trustworthy, since it converges to the analytical free energy corresponding to a naked ion for $z = -3 \text{ nm}$, the solid black line in Fig. 3b, whereas the umbrella method does not. This shows that the WHAM method produces a free energy landscape that does not reflect the expected limiting behavior, which is the reason why we only show it in the Supplement. We added a discussion on the limitations of the umbrella sampling method in the Supplement.

The authors should be commended for performing two sets of umbrella sampling simulations starting from different initial states, i.e., the hydrated ion in water (forward, solid green line) and the bare ion in vapor (reverse, dashed green line). The authors show that free energy profiles obtained from the two sets of simulations do not agree with one another due to the hysteresis associated with ion hydration and water finger formation; these results highlight the challenges in estimating $U(z)$.

We totally agree with this statement. Ion hydration in the vapor phase is an extremely slow process which is only obtained after about 100 ns simulations, since the water molecules equilibrate with the vapor phase, which is very dilute. Note that the ion hydration in the vapor phase is missed by both TI and umbrella-sampling methods, therefore we estimate the free energy of the hydrated ion in vapor from the hydration-number distribution and from two analytical models.

The true $U(z)$ is likely to lie between the forward and reverse umbrella sampling $U(z)$ profiles, i.e., the solid and broken green lines in Fig. S8, suggesting that the reversible work of evaporating the Cl is substantially lower than that suggested by the cubic fit the TI free energy profile.

As argued above, the umbrella free energy is not reliable since it disagrees with the limiting free energy of a naked ion, denoted by a black solid line in Fig. S8b. This is why, we fit our cubic model to the TI data in the distance range $-2.0 \text{ nm} < z < 0.4 \text{ nm}$ with the constraint that the cubic model reaches the ion-hydration corrected image-charge energy (broken curve in Fig. 3b).

How sensitive are overall conclusions to the choice of $U(z)$, e.g., what would the predicted atmospheric chloride concentration be if forward (and reverse) umbrella sampling results were used to estimate it instead?

For a decrease of the barrier energy by $40 k_B T$, as judged from Fig. S8b, using the Boltzmann factor the rate would increase by an enormous factor of $e^{40} \sim 10^{17}$, which would indeed change our conclusions. However, as explained above, the umbrella sampling results do not converge to the correct asymptotic limit and thus cannot be used to estimate the ion-solvation free energy.

Finally, the authors may also be able to reduce hysteresis in the umbrella sampling simulations by using a hydrated ion (with 7 water molecules rather than a bare ion) to estimate the reverse $U(z)$ profiles.

We thank the reviewer for this interesting suggestion. However, for reasons explained above, the TI method seems to be the more reliable method to estimate the free energy in the distance range $-2.0 \text{ nm} < z < 0.4 \text{ nm}$, as explained in the SI.

2) I could be mistaken, but a back-of-the-envelope calculation suggests that for the estimated $U(z)$, the value of $\sqrt{2 \pi kT / U''_{\text{min}}}$ is much smaller than the simulation box size. The authors are encouraged to justify their use of eq. 1 with $L=0$ to obtain the transition state diffusivity (in Fig. 5).

The limit $L=0$ correctly describes the potential felt by the chloride ion when we apply a constant force that pushes the chloride ion into the vapor phase, as seen in Fig. 4f. This scenario is used to extract the transition state diffusivity D_{tr} from the data when we apply such a force. Only when one wants to estimate the chloride flux from a bulk phase does L correspond to the system size. We added clarifying comments in the revised version.

3) As the authors (and others) have shown, the water finger plays a central role in the ion evaporation process. Should we expect a coupling between the distortion of the water-vapor interface in the vicinity of the water finger and the capillary wave spectrum of the interface far from the water finger? If so, might the free energy profiles determined by the authors (and therefore, the barrier to ion evaporation) depend on the cross-section of the simulation box?

The referee raises an interesting and subtle point. We cannot exclude finite size effects that would depend on the lateral system size, although our choice of 9024 water molecules and a lateral area of $5.1 \text{ nm} \times 5.4 \text{ nm}$ and a slab thickness of 10 nm seems sufficient. This we say because we obtain good agreement of the water vapor pressure between simulations and experimental values, though it cannot be excluded that ion evaporation processes are subject to stronger finite-size effects. The system size used definitely marks our simulation capacity. We added cautious remarks on possible finite-size effects in the paper.

4) For the water number distribution shown in Fig. 2h, do the mean and the standard deviation become independent of z below -7 nm ? The authors estimate the free energetics of ion hydration assuming that this distribution is Gaussian. The authors are encouraged to explain why this is a reasonable assumption. A plot of $-\log P$ (perhaps as inset or in SI) may also help highlight that the fluctuations are Gaussian not just near the mean, but also in the tails.

We thank the reviewer for these comments. Because of the scarcity of our numerical data for the ion hydration distribution in the vapor phase, we cannot really prove that the distribution is Gaussian. We would like to note that ion hydration in the vapor phase is an extremely slow process, which is probably the reason why it has not been observed before, neither in simulations nor in experiments. The numerical problems of simulating the ion hydration process in vapor led us to develop two analytic models for the hydration process, which are explained in the paper. Both models confirm that monovalent ions should be hydrated in the vapor phase and give a hydration shell thickness and a hydration free energy that is consistent with our simulation estimate. We added some cautious remarks along these lines in the paper. Also, for a separation of -7 nm from the air-water interface, the image-charge interaction potential has reached a constant value, as seen in Fig. S6a in the Supplement, since this is the only interaction with the interface, we expect a distance of -7 nm to correspond to vapor bulk.

Minor Points / Typos:

- 1) Page 4: Fig. 2 g, h  Fig. 1 g, h.
- 2) Caption of Fig. 4: Two instances of $F_{pull} = 44.3$  44.1 to be consistent with the text.
- 3) The cubic fits (red line) in Fig. 3b and Fig. S8b appear to be different.

We thank the reviewer for pointing out these typos, helping us to improve our paper.

Reviewer #2

“Evaporation of Nano-Hydrated Ions from Aqueous Solution” by Netz and co-workers studies the transfer of a chloride ion from the aqueous phase to an adjacent vapor phase. The phenomenological finding of this study, that chloride ions do not spontaneously ‘evaporate’ at any appreciable rate and therefore do not contribute significantly to atmospheric chloride, seems rather obvious. A preliminary literature survey of the phase transfer thermodynamics and mechanism of chloride should reveal this outcome, e.g. Reference 17,18, and other works not included in the main text. Comparisons to related work in the academic literature are also lacking. For these reasons, the motivations and background of this work as presented by the authors are not strong. Revisiting atmospheric Cl at the end of the manuscript with related flux calculations seems forced and whimsical.

However, these simulations are interesting as an academic study of this rare event and may serve as an interesting companion piece to earlier work in the field. The simulations and analyses appear sound but again would benefit from comparison to existing work. The novelty of the results of these calculations is very low and will not significantly influence thinking in the field. Regarding potential impact, I would recommend publication in a more specialized journal.

I believe that addressing the following concerns would strengthen this work, whether resubmission to this journal is permitted or if submitted elsewhere:

- 1) *The construction of the introduction and main findings should be reconsidered, particularly considering the comments above. A purely academic investigation of Cl water/vapor transfer seems far more compelling than suggesting that it may be a significant contributor to atmospheric Cl.*

We do not fully agree with this interpretation of our manuscript. The most important and in fact very surprising finding of our work is that a monovalent ion is hydrated in the vapor phase, which in our view constitutes a major change of our understanding of vaporized ions. Consequently, the ion solvation free energy decreases dramatically by $26 k_B T$, which in turn means that the estimated evaporation rate increases by an enormous factor of $e^{26} = 10^{11}$! That the resulting chloride evaporation rate from the seas is still not high enough to explain the current atmospheric chloride concentration is in our opinion far from obvious, since it depends on a careful estimation of the ion hydration free energy in the vapor phase and the transition state diffusion constant. We also would not call our finding “phenomenological”, as we carefully estimate all microscopic parameters that appear in the reaction rate theory for the evaporation process. We note in passing that to the best of our knowledge, our paper is the first to estimate the ion hydration free energy in the vapor phase and the first to determine the transition state diffusivity of an ion, so it is the first time that an ion evaporation rate can be accurately determined.

The discussion of the kinetic process that establishes the atmospheric chloride concentration is definitely not the main point of our paper, but we see no reason to remove this discussion from our paper and with the foregoing arguments do not agree that this discussion is “whimsical”.

In order to show that this is not the main point of our paper, we have removed the sentence on the atmospheric chloride concentration from the abstract.

We note that reference 17 mentioned by the referee did not explicitly consider the hydrated ion equilibrium state in the vapor phase, presumably because a small water droplet was simulated. Reference 18 studied an ion at the water-oil interface, which is different from the liquid-vapor water interface.

- 2) *The ‘water finger’ accompanying [Cl⁻] ion transfer phenomena was described nearly 30 years ago and should be cited, e.g. Science 1993, 261 (5128), 1558–1560. The authors showcase this mechanism in several figures and the main text quite often. Also regarding the mechanism, the authors should compare this event to the evaporation of water, which was shown to occur one water molecule at-a-time: Phys. Rev. Lett. 2015, 115 (23), 236102.*

We thank the reviewer for pointing out these two very relevant references, which we missed in our manuscript. The Science paper from 1993 indeed introduces the water finger and we refer to this important paper several times in the revised version of our paper. The PRL from 2015, which we cite in the new version of our paper, nicely discusses the microscopic mechanism of how a water molecule evaporates from the liquid-vapor water phase. This paper does not address the free energetics of the evaporation, which is not needed as the evaporation free energy of a water molecule is determined by the water vapor pressure. Also, it is clear that water molecules in the vapor phase are not hydrated but rather present as isolated water molecules, so the scenario of the evaporation of a water molecule is much simpler than the evaporation of an ion into the vapor phase. We added a short discussion of the 2015 PRL in the Introduction.

- 3) *The Methods section in the main text contains very little information regarding the simulations. The basic class of the simulations, fixed-charge classical MD, should be clear to the reader. In its current form, the simulation approach is not clear, with most information relegated to the SI. This could be remedied with minimal impact on word count by simply mentioning the water and Cl⁻ models in the main text.*

We added additional details on the simulation methods in the main text.

- 4) *Related to (3), the authors do not consider AIMD or polarizable models, which would probably affect the results dramatically. In the current formalism, charge scaling could also be considered to compensate for the lack of polarizability, e.g. J. Phys. Chem. Lett. 2019, 7531–7536 and maybe an interesting addition to this work.*

These are all very valid comments. We recently optimized non-polarizable parameters for ions (Loche et al., J. Phys. Chem. B 125, 2021, 8581–8587) in non-polarizable water models with respect to experimental ion solvation free energies and ion activity coefficients and showed that other experimental observables like mass density, conductivity and the dielectric constant are faithfully reproduced over the entire concentration range up to 4M. This means that explicit polarizabilities are not needed to reproduce bulk behavior in force-field simulations. These optimized parameters are close to the ones used in the present work, as we note in the revised version.

At interfaces, polarizability effects are potentially more relevant. In previous work, we showed that rigorously optimized non-polarizable force fields accurately reproduce ionic interfacial properties in agreement with experiments and ab initio simulations (Mamatkulov et al., Angewandte Chemie 56, 2017, 15846–15851). Nevertheless, the main problem at interfaces is that the water dipole moment M is reduced compared to the bulk value, as shown below in Fig. 1c, where we compare force-field and ab initio simulation results for a water-slab system. The problem is that ab initio simulations are unfeasible when it comes to predicting the rare events, we are study in our paper, since the maximal ab initio simulation time for 256 water molecules is about 200 ps, whereas we have simulated 9024 water molecules for many microseconds in our work. Polarizable ion and

water models also substantially increase the numerical burden; besides, we are not aware of polarizable models that would accurately reproduce experimental ion solvation free energies and water vapor pressure at the same time, which is needed for our work. Charge scaling is an attractive idea in mixed systems but does not really solve the problem of the changing water dipole moment M at an interface. It is also not needed since our chloride force field reproduces all bulk properties very accurately, as explained above and mentioned in the revised version of our paper, where we also include the reference suggested by the referee. In essence, our methodology was carefully chosen as a compromise between numerical efficiency and accuracy.

Fig. 1: Comparison of force-field (FF) and ab initio (DFT) MD simulations of a water slab consisting of 256 water molecules. The mass density profiles in b) compare rather well. The water dipole moment profile from the ab initio MD simulations in c) shows a significant reduction at the interface, the water dipole moment in vapor is indicated by a broken horizontal line (M. Becker and R.R. Netz, unpublished results).

- 5) *The spontaneous breaking and reformation of the “water finger” in liquid/liquid Cl-transfer has been previously reported, J. Chem. Phys. 2016, 145 (1), 014701, differences in water finger lengths and dynamics in liquid/liquid versus liquid/vapor may provide more mechanistic insight.*

We thank the reviewer for pointing out this reference, which we gratefully included in the revised version of our manuscript.

- 6) *The results shown in 3b are interesting and show different approaches (TI & a simple image charge) arrive at the same state function at the beginning and end with mechanistic deviation seen in the free energy profiles. However, the corresponding discussion in the text is difficult to parse and should be revised. Also, why is the TI free energy profile still increasing at $z = -3$ nm? Is there an endpoint plateau?*

We clarified the section explaining the results of Fig. 3b in our revised manuscript. The plateau in the image-charge repulsion is reached for long distances ($z < -10$ nm), as we show in Fig.S6a in the SI.

- 7) *Animations should be included as SI (no youtube links.)*

We additionally provide all animations in the SI.

Reviewer #3

In this work, the authors have presented theoretical studies of evaporation of a chloride ion from a water surface by carrying out four different equilibrium and non-equilibrium molecular dynamics simulations. The work reveals the important result that the chloride ion evaporates in a nano-hydrated form which substantially reduces the free energy barrier for evaporation and accelerates the rate of evaporation. Despite this favorable route for evaporation of chloride ions in nano-hydrated form, the predicted concentration of chloride ions in the atmosphere is still much lower than what is experimentally reported to be present over the ocean. Thus, the present study also suggests that a major part of chloride ions in the atmosphere come not through normal evaporation, but non-equilibrium processes caused by oceanic waves and wind. The current results are important and, in my opinion, merits publication in Communications Chemistry. The following point, however, may be noted.

The current results are for the SPCE model of water and charged Lennard-Jones model of ions. Thus, none of the solvents and ions have polarizability. Earlier studies have shown that polarizability can be important for the surface behavior of anions. This point may be discussed.

Indeed, early non-polarizable anion force fields did not correctly reproduce the weak tendency of anions to adsorb onto the air-water interface, while polarizable force fields did. Later, thermodynamically optimized force fields were shown to correctly and quantitatively reproduce the experimental interfacial excesses of various ions, including the tricky ions H_3O^+ and OH^- , as shown in Mamatkulov et al., *Angewandte Chemie* 56, 2017, 15846–15851. Nevertheless, as we demonstrate in the above Fig. 1 by ab initio simulations, polarizability effects are clearly present at vapor-liquid water interfaces. However, since our ionic force fields are optimized to reproduce the solvation free energy of ions, we presume that polarizability effects will not grossly change our conclusions. We added some cautious remarks and explanations into the paper.

What could be the effects of finite concentration? For example, NaCl concentration in ocean water is about 0.6 M. It would be good to discuss this concentration issue as well.

This is an important point which indeed was not discussed in the submitted version of our paper. The main effect of a finite concentration is on the solvation free energy of an ion. We write the chemical potential of an ion as

$$\mu = \mu_0 + k_B T \ln(c\gamma) = \mu_0 + k_B T \ln(c) + k_B T \ln(\gamma)$$

where μ_0 is the chemical potential in the zero-concentration limit, which is the solvation free energy, c is the ion concentration in some dimensionless units and γ is the concentration-dependent activity coefficient. The ideal contribution to the chemical potential, the $k_B T \ln(c)$ term, is explicitly taken into account in our expression for the evaporation flux and produces the linear concentration dependence in Eq. (8) in our paper. The experimental value for the activity coefficient of a NaCl solution at 0.6 M is about $\gamma = 0.73$, see for example our paper dos Santos et al, *J. Chem. Phys.* 153, 2020, 034103. Therefore, the salt concentration-dependent correction to the ion solvation free energy amounts to $k_B T \ln(0.73) = -0.32 k_B T$ and therefore can be neglected. We added a discussion in our paper.

The other possible correction could come from the salt-concentration dependence of the viscosity, but since what matters is really the viscosity of the water in the transition state, when the finger is about to rupture, we would assume that the finite salt concentration in the bulk is not relevant to first order (although it would be interesting to study this in more detail).

Otherwise, this is an important piece of work. The paper is written well and I support its publication.

We thank the referee for the final remark and hope that the revised version of our paper will be accepted for publication.

Yours sincerely,
the authors

REVIEWERS' COMMENTS:

Reviewer #1 (Remarks to the Author):

In revising their manuscript, the authors have satisfactorily addressed the points raised by this reviewer. I am therefore happy to recommend publication of this interesting manuscript.

Reviewer #2 (Remarks to the Author):

Comments inserted into the authors' rebuttal text, attached as pdf.

Reviewer #3 (Remarks to the Author):

I have gone through the revised version. I am satisfied with the revisions made by the authors. In my opinion, the revised manuscript can be accepted for publication in Communications Chemistry.

Reviewer 2 comments in green text.

Thank you very much for the reports. We thank the three reviewers for their constructive criticism and the support of our manuscript. We highlight all changes in the manuscript in red, except minor modifications. In the following, we reproduce the referee reports in full and reply to the referees' suggestions point by point.

Reviewer #1

In this manuscript, the authors estimate the rate at which a chloride ion evaporates from liquid water into saturated water vapor using molecular simulations and enhanced sampling calculations.

The authors find that chloride ion does not lose all its hydration waters upon evaporation but retains roughly 7 waters in the vapor phase. Consistent with previous studies, the authors show that ion evaporation is mediated by the formation of a water finger, and further show that the water finger becomes unstable roughly 2.8 nm away from the water- vapor interface.

To predict the overall rate of ion evaporation, the authors estimate the free energetic barrier for evaporation as well as the effective diffusivity of the chloride ion in the transition state. They find that the latter is roughly 6 times the diffusivity of the ion in bulk water.

The authors further find that the estimated kinetics of chloride evaporation (from a quiescent interface) are too slow to rationalize the chloride evaporation flux from the earth's oceans. They speculate that spray formation due to wind and oceanic waves must be the dominant contributors to atmospheric chloride concentration.

1) The noise in the TI results shown in Fig. 3b for z below -1 nm (and the corresponding deviation from the smooth cubic fit) suggests the presence of substantial error in $U(z)$, and even more so in the corresponding forces that are estimated in the simulations. Given that these errors influence the estimated ion evaporation barrier, the authors are encouraged to include error bars.

Assuming that there is overlap between the umbrella sampling windows (for the free energy profiles reported in Fig. S8b), I am inclined to trust the authors' umbrella sampling results more than the TI results.

We thank the reviewer for pointing out that the discussion of errors was incomplete in the previous version of the manuscript. Since this discussion is important to understand why we prefer our TI results over our umbrella sampling results, we added details on our error estimates of both simulation results in the SI and also mention the error of the TI method briefly in the main text. In short: The estimated error of the TI results, calculated via error propagation from the individual errors of each integration step, is about ~ 1 k!T and thus substantially smaller than the mean deviations from the fit function in Fig. 3b, which is about ~ 5 k!T in the range -2.0 nm $< z < -1.0$ nm. We mention that the estimated error is smaller than the symbol size in the revised version in the caption of Fig. 3b and therefore error bars would not be visible in Fig. 3b. The difference between the calculated errors and the visible deviations from the fit function most likely comes from the slow dynamics of the water-finger fluctuations,

which occurs over more than 100 ns, as shown in Fig. 2g. The estimated error of the data obtained from umbrella sampling is about ~ 5 k!T and is in the revised version shown in Fig. S8b in the Supplement, where we now present the individual data points from the WHAM analysis and not a line through the data, as done in the previous version. We used umbrella potentials with a spacing of 0.2 nm that sufficiently overlap (note that the data points in Fig. S8b result from postprocessing via the WHAM analysis and have a much smaller separation). However, when the water finger is present (for $z < -1.5$ nm), the sampling time of 5 ns per potential in the umbrella sampling protocol is rather short. We therefore trust the TI data more, where we spent 50 ns per integration step in the distance range where the water finger is present, which results for 20 integration steps in a total of 1 μ s simulation time per data point. Most importantly, the TI data is more trustworthy, since it converges to the analytical free energy corresponding to a naked ion for $z = -3$ nm, the solid black line in Fig. 3b, whereas the umbrella method does not. This shows that the WHAM method produces a free energy landscape that does not reflect the expected limiting behavior, which is the reason why we only show it in the Supplement. We added a discussion on the limitations of the umbrella sampling method in the Supplement.

The authors should be commended for performing two sets of umbrella sampling simulations starting from different initial states, i.e., the hydrated ion in water (forward, solid green line) and the bare ion in vapor (reverse, dashed green line). The authors show that free energy profiles obtained from the two sets of simulations do not agree with one another due to the hysteresis associated with ion hydration and water finger formation; these results highlight the challenges in estimating $U(z)$.

We totally agree with this statement. Ion hydration in the vapor phase is an extremely slow process which is only obtained after about 100 ns simulations, since the water molecules equilibrate with the vapor phase, which is very dilute. Note that the ion hydration in the vapor phase is missed by both TI and umbrella-sampling methods, therefore we estimate the free energy of the hydrated ion in vapor from the hydration-number distribution and from two analytical models.

The true $U(z)$ is likely to lie in between the forward and reverse umbrella sampling $U(z)$ profiles, i.e., the solid and broken green lines in Fig. S8, suggesting that the reversible work of evaporating the Cl is substantially lower than that suggested by the cubic fit the TI free energy profile.

As argued above, the umbrella free energy is not reliable since it disagrees with the limiting free energy of a naked ion, denoted by a black solid line in Fig. S8b.

A few questions related to Reviewer 1's comments: Why would the "the limiting free energy of a naked ion" be the limit for umbrella sampling? Shouldn't the limiting [i.e. 'evaporated state'] be the ion with [equilibrium] co-transferred water? Additionally, in any case why would ΔU be greater for the hydrated ion (water to vapor) than the bare ion (vapor to water) in the umbrella sampling results? The authors' insights regarding TI vs. US would be very interesting here.

This is why, we fit our cubic model to the TI data in the distance range $-2.0 \text{ nm} < z < 0.4 \text{ nm}$ with the constraint that the cubic model reaches the ion-hydration corrected image-charge energy (broken curve in Fig. 3b).

The actual TI data significantly overshoots the 'shifted analytical curve' and the TI "fit" at $z < 2.3 \text{ nm}$, as described (and well explained) by the authors. Would this TI data follow the solid black curve when $z < -3 \text{ nm}$?

How sensitive are overall conclusions to the choice of $U(z)$, e.g., what would the predicted atmospheric chloride concentration be if forward (and reverse) umbrella sampling results were used to estimate it instead?

For a decrease of the barrier energy by 40 kBT, as judged from Fig. S8b, using the Boltzmann factor the rate would increase by an enormous factor of $e^{\sim 10}$, which would indeed change our conclusions. However, as explained above, the umbrella sampling results do not converge to the correct asymptotic limit and thus cannot be used to estimate the ion-solvation free energy.

Finally, the authors may also be able to reduce hysteresis in the umbrella sampling simulations by using a hydrated ion (with 7 water molecules rather than a bare ion) to estimate the reverse $U(z)$ profiles.

We thank the reviewer for this interesting suggestion. However, for reasons explained above, the TI method seems to be the more reliable method to estimate the free energy in the distance range $-2.0 \text{ nm} < z < 0.4 \text{ nm}$, as explained in the SI.

The approach suggested by Reviewer 1 was successful in previously published liquid/liquid simulations and showed little hysteresis. I presume this would also be the case for liquid/vapor.

2) I could be mistaken, but a back-of-the-envelope calculation suggests that for the estimated $U(z)$, the value of $\sqrt{2 \pi kT / U''_{\text{min}}}$ is much smaller than the simulation box size. The authors are encouraged to justify their use of eq. 1 with $L=0$ to obtain the transition state diffusivity (in Fig. 5).

The limit $L=0$ correctly describes the potential felt by the chloride ion when we apply a constant force that pushes the chloride ion into the vapor phase, as seen in Fig. 4f. This scenario is used to extract the transition state diffusivity D_{tr} from the data when we apply such a force. Only when one wants to estimate the chloride flux from a bulk phase does L correspond to the system size. We added clarifying comments in the revised version.

3) As the authors (and others) have shown, the water finger plays a central role in the ion evaporation process. Should we expect a coupling between the distortion of the water-vapor interface in the vicinity of the water finger and the capillary wave spectrum of the interface far

from the water finger? If so, might the free energy profiles determined by the authors (and therefore, the barrier to ion evaporation) depend on the cross-section of the simulation box?

The referee raises an interesting and subtle point. We cannot exclude finite size effects that would depend on the lateral system size, although our choice of 9024 water molecules and a lateral area of 5.1 nm x 5.4 nm and a slab thickness of 10 nm seems sufficient. This we say because we obtain good agreement of the water vapor pressure between simulations and experimental values, though it cannot be excluded that ion evaporation processes are subject to stronger finite-size effects. The system size used definitely marks our simulation capacity. We added cautious remarks on possible finite-size effects in the paper.

4) For the water number distribution shown in Fig. 2h, do the mean and the standard deviation become independent of z below -7 nm? The authors estimate the free energetics of ion hydration assuming that this distribution is Gaussian. The authors are encouraged to explain why this is a reasonable assumption. A plot of $-\log P$ (perhaps as inset or in SI) may also help highlight that the fluctuations are Gaussian not just near the mean, but also in the tails.

We thank the reviewer for these comments. Because of the scarcity of our numerical data for the ion hydration distribution in the vapor phase, we cannot really prove that the distribution is Gaussian. We would like to note that ion hydration in the vapor phase is an extremely slow process, which is probably the reason why it has not been observed before, neither in simulations nor in experiments. The numerical problems of simulating the ion hydration process in vapor led us to develop two analytic models for the hydration process, which are explained in the paper. Both models confirm that monovalent ions should be hydrated in the vapor phase and give a hydration shell thickness and a hydration free energy that is consistent with our simulation estimate. We added some cautious remarks along these lines in the paper. Also, for a separation of -7 nm from the air-water interface, the image-charge interaction potential has reached a constant value, as seen in Fig. S6a in the Supplement, since this is the only interaction with the interface, we expect a distance of -7 nm to correspond to vapor bulk.

Simulating a $\text{Cl}(\text{H}_2\text{O})_n^-$ complex in vacuum could be a computationally inexpensive approach toward describing the evaporated hydrated ion far from the liquid/vapor interface. A limiting case for the $z < -2.x$ region, where the liquid/ion complex interaction is very small.

Minor Points / Typos:

1) Page 4: Fig. 2 g, h  Fig. 1 g, h.

2) Caption of Fig. 4: Two instances of $F_{\text{pull}} = 44.3$  44.1 to be consistent with the text. 3) The cubic fits (red line) in Fig. 3b and Fig. S8b appear to be different.

We thank the reviewer for pointing out these typos, helping us to improve our paper.

Reviewer #2

“Evaporation of Nano-Hydrated Ions from Aqueous Solution” by Netz and co-workers studies the transfer of a chloride ion from the aqueous phase to an adjacent vapor phase. The phenomenological finding of this study, that chloride ions do not spontaneously ‘evaporate’ at any appreciable rate and therefore do not contribute significantly to atmospheric chloride, seems rather obvious. A preliminary literature survey of the phase transfer thermodynamics and mechanism of chloride should reveal this outcome, e.g. Reference 17,18, and other works not included in the main text. Comparisons to related work in the academic literature are also lacking. For these reasons, the motivations and background of this work as presented by the authors are not strong. Revisiting atmospheric Cl at the end of the manuscript with related flux calculations seems forced and whimsical.

However, these simulations are interesting as an academic study of this rare event and may serve as an interesting companion piece to earlier work in the field. The simulations and analyses appear sound but again would benefit from comparison to existing work. The novelty of the results of these calculations is very low and will not significantly influence thinking in the field. Regarding potential impact, I would recommend publication in a more specialized journal. I believe that addressing the following concerns would strengthen this work, whether resubmission to this journal is permitted or if submitted elsewhere:

1) The construction of the introduction and main findings should be reconsidered, particularly considering the comments above. A purely academic investigation of Cl water/vapor transfer seems far more compelling than suggesting that it may be a significant contributor to atmospheric Cl.

We do not fully agree with this interpretation of our manuscript. The most important and in fact very surprising finding of our work is that a monovalent ion is hydrated in the vapor phase, which in our view constitutes a major change of our understanding of vaporized ions. Consequently, the ion solvation free energy decreases dramatically by 26 kBT, which in turn means that the estimated evaporation rate increases by an enormous factor of $e^{26} = 10^{11}$! That the resulting chloride evaporation rate from the seas is still not high enough to explain the current atmospheric chloride concentration is in our opinion far from obvious, since it depends on a careful estimation of the ion hydration free energy in the vapor phase and the transition state diffusion constant. We also would not call our finding “phenomenological”, as we carefully estimate all microscopic parameters that appear in the reaction rate theory for the evaporation process. We note in passing that to the best of our knowledge, our paper is the first to estimate the ion hydration free energy in the vapor phase and the first to determine the transition state diffusivity of an ion, so it is the first time that an ion evaporation rate can be accurately determined.

The discussion of the kinetic process that establishes the atmospheric chloride concentration is definitely not the main point of our paper, but we see no reason to remove this discussion from our paper and with the foregoing arguments do not agree that this discussion is “whimsical”.

In order to show that this is not the main point of our paper, we have removed the sentence on the atmospheric chloride concentration from the abstract.

We note that reference 17 mentioned by the referee did not explicitly consider the hydrated ion equilibrium state in the vapor phase, presumably because a small water droplet was

simulated. Reference 18 studied an ion at the water-oil interface, which is different from the liquid-vapor water interface.

2) The 'water finger' accompanying [Cl⁻] ion transfer phenomena was described nearly 30 years ago and should be cited, e.g. Science 1993, 261 (5128), 1558–1560. The authors showcase this mechanism in several figures and the main text quite often. Also regarding the mechanism, the authors should compare this event to the evaporation of water, which was shown to occur one water molecule at-a-time: Phys. Rev. Lett. 2015, 115 (23), 236102.

We thank the reviewer for pointing out these two very relevant references, which we missed in our manuscript. The Science paper from 1993 indeed introduces the water finger and we refer to this important paper several times in the revised version of our paper. The PRL from 2015, which we cite in the new version of our paper, nicely discusses the microscopic mechanism of how a water molecule evaporates from the liquid-vapor water phase. This paper does not address the free energetics of the evaporation, which is not needed as the evaporation free energy of a water molecule is determined by the water vapor pressure. Also, it is clear that water molecules in the vapor phase are not hydrated but rather present as isolated water molecules, so the scenario of the evaporation of a water molecule is much simpler than the evaporation of an ion into the vapor phase. We added a short discussion of the 2015 PRL in the Introduction.

The authors have satisfactorily addressed (2). That water transfers to the vapor phase one-at-a-time and transfers Cl⁻ with significant co-transfer of solvating water suggests interesting spaces in between, Δq and $\Delta \text{species}$.

3) The Methods section in the main text contains very little information regarding the simulations. The basic class of the simulations, fixed-charge classical MD, should be clear to the reader. In its current form, the simulation approach is not clear, with most information relegated to the SI. This could be remedied with minimal impact on word count by simply mentioning the water and Cl⁻ models in the main text.

We added additional details on the simulation methods in the main text.

Thank you, the authors have satisfactorily addressed (3).

4) Related to (3), the authors do not consider AIMD or polarizable models, which would probably affect the results dramatically. In the current formalism, charge scaling could also be considered to compensate for the lack of polarizability, e.g. J. Phys. Chem. Lett. 2019, 7531–7536 and maybe an interesting addition to this work.

These are all very valid comments. We recently optimized non-polarizable parameters for ions (Loche et al., J. Phys. Chem. B 125, 2021, 8581–8587) in non-polarizable water models with respect to experimental ion solvation free energies and ion activity coefficients and showed that other experimental observables like mass density, conductivity and the dielectric constant

are faithfully reproduced over the entire concentration range up to 4M. This means that explicit polarizabilities are not needed to reproduce bulk behavior in force-field simulations. These optimized parameters are close to the ones used in the present work, as we note in the revised version.

At interfaces, polarizability effects are potentially more relevant. In previous work, we showed that rigorously optimized non-polarizable force fields accurately reproduce ionic interfacial properties in agreement with experiments and *ab initio* simulations (Mamatkulov et al., *Angewandte Chemie* 56, 2017, 15846–15851). Nevertheless, the main problem at interfaces is that the water dipole moment M is reduced compared to the bulk value, as shown below in Fig. 1c, where we compare force-field and *ab initio* simulation results for a water-slab system. The problem is that *ab initio* simulations are unfeasible when it comes to predicting the rare events, we are study in our paper, since the maximal *ab initio* simulation time for 256 water molecules is about 200 ps, whereas we have simulated 9024 water molecules for many microseconds in our work. Polarizable ion and water models also substantially increase the numerical burden; besides, we are not aware of polarizable models that would accurately reproduce experimental ion solvation free energies and water vapor pressure at the same time, which is needed for our work. Charge scaling is an attractive idea in mixed systems but does not really solve the problem of the changing water dipole moment M at an interface. It is also not needed since our chloride force field reproduces all bulk properties very accurately, as explained above and mentioned in the revised version of our paper, where we also include the reference suggested by the referee. In essence, our methodology was carefully chosen as a compromise between numerical efficiency and accuracy.

Fig. 1: Comparison of force-field (FF) and ab initio (DFT) MD simulations of a water slab consisting of 256 water molecules. The mass density profiles in b) compare rather well. The water dipole moment profile from the ab initio MD simulations in c) shows a significant reduction at the interface, the water dipole moment in vapor is indicated by a broken horizontal line (M. Becker and R.R. Netz, unpublished results).

A similar comment was given by Reviewer 3. Thank you for this addition. I agree with the authors' force field selection and interpretation. Commentary that justifies a model's selection and its limitations is important and significantly strengthens the Methods section. The authors have satisfactorily addressed (4).

5) The spontaneous breaking and reformation of the "water finger" in liquid/liquid Cl- transfer has been previously reported, J. Chem. Phys. 2016, 145 (1), 014701, differences in water finger lengths and dynamics in liquid/liquid versus liquid/vapor may provide more mechanistic insight.

We thank the reviewer for pointing out this reference, which we gratefully included it the revised version of our manuscript.

The authors have satisfactorily addressed (5).

6) The results shown in 3b are interesting and show different approaches (TI & a simple image charge) arrive at the same state function at the beginning and end with mechanistic deviation

seen in the free energy profiles. However, the corresponding discussion in the text is difficult to parse and should be revised. Also, why is the TI free energy profile still increasing at $z = -3$ nm? Is there an endpoint plateau?

We clarified the section explaining the results of Fig. 3b in our revised manuscript. The plateau in the image-charge repulsion is reached for long distances ($z < -10$ nm), as we show in Fig.S6a in the SI.

This clarification is very much appreciated. The TI approach and justification for omitting data in the cubic fit is justified in the text but as mentioned, was difficult for me to parse as presented. I would suggest an attempt to clarify this aspect of the writing but leave the decision of whether to revise to the authors.

Additionally, the question I inserted into Reviewer 1's comments: Do the authors think that the TI data would follow the solid black curve when $z < -3$ nm? That is, would the TI data 'kink' at this point?

The authors have satisfactorily addressed (6).

7) Animations should be included as SI (no youtube links.)

We additionally provide all animations in the SI.

Thank you, these animations are very well made and sure to inspire interested readers.

Related to comments in (1) and (4), I believe that the transfer of Cl⁻ from the aqueous phase into vacuum (vapor phase) fits into a larger body of work. Transferring an ion or a water molecule from the aqueous phase to vacuum or to an adjacent immiscible liquid phase is an interesting set of molecular mechanics simulations where electrostatics appear to govern the co-transfer (partial or none) of the transferring species' hydration shell.

I support the publication of this manuscript in its revised form.

Reviewer #3

In this work, the authors have presented theoretical studies of evaporation of a chloride ion from a water surface by carrying out four different equilibrium and non-equilibrium molecular dynamics simulations. The work reveals the important result that the chloride ion evaporates in a nano-hydrated form which substantially reduces the free energy barrier for evaporation and accelerates the rate of evaporation. Despite this favorable route for evaporation of chloride ions in nano-hydrated form, the predicted concentration of chloride ions in the atmosphere is still much lower than what is experimentally reported to be present over the ocean. Thus, the present study also suggests that a major part of chloride ions in the atmosphere come not through normal evaporation, but non-equilibrium processes caused by oceanic waves and wind.

The current results are important and, in my opinion, merits publication in Communications Chemistry. The following point, however, may be noted.

The current results are for the SPCE model of water and charged Lennard-Jones model of ions. Thus, none of the solvents and ions have polarizability. Earlier studies have shown that polarizability can be important for the surface behavior of anions. This point may be discussed.

Indeed, early non-polarizable anion force fields did not correctly reproduce the weak tendency of anions to adsorb onto the air-water interface, while polarizable force fields did. Later, thermodynamically optimized force fields were shown to correctly and quantitatively reproduce the experimental interfacial excesses of various ions, including the tricky ions H₃O⁺ and OH⁻, as shown in Mamatkulov et al., *Angewandte Chemie* 56, 2017, 15846–15851. Nevertheless, as we demonstrate in the above Fig. 1 by ab initio simulations, polarizability effects are clearly present at vapor-liquid water interfaces. However, since our ionic force fields are optimized to reproduce the solvation free energy of ions, we presume that polarizability effects will not grossly change our conclusions. We added some cautious remarks and explanations into the paper.

What could be the effects of finite concentration? For example, NaCl concentration in ocean water is about 0.6 M. It would be good to discuss this concentration issue as well.

This is an important point which indeed was not discussed in the submitted version of our paper. The main effect of a finite concentration is on the solvation free energy of an ion. We write the chemical potential of an ion as

$$\mu = \mu^\# + kT \ln(c\gamma) = \mu^\# + kT \ln(c) + kT \ln(\gamma)$$

where $\mu^\#$ is the chemical potential in the zero-concentration limit, which is the solvation free energy, c is the ion concentration in some dimensionless units and γ is the concentration-dependent activity coefficient. The ideal contribution to the chemical potential, the $kT \ln(c)$ term, is explicitly taken into account in our expression for the evaporation flux and produces the linear concentration dependence in Eq. (8) in our paper. The experimental value for the activity coefficient of a NaCl solution at 0.6 M is about $\gamma = 0.73$, see for example our paper dos Santos et al, *J. Chem. Phys.* 153, 2020, 034103. Therefore, the salt concentration-dependent correction to the ion solvation free energy amounts to $kT \ln(0.73) = -0.32 kT$ and therefore can be neglected. We added a discussion in our paper.

The other possible correction could come from the salt-concentration dependence of the viscosity, but since what matters is really the viscosity of the water in the transition state, when the finger is about to rupture, we would assume that the finite salt concentration in the bulk is not relevant to first order (although it would be interesting to study this in more detail).

Otherwise, this is an important piece of work. The paper is written well and I support its publication.

We thank the referee for the final remark and hope that the revised version of our paper will be accepted for publication.

Yours sincerely,
the authors

Prof. Roland Netz

Phone: +49 30 838 55 737

Fax: +49 30 838 53741

Mail: metz@physik.fu-berlin.de

Web: <https://www.physik.fu-berlin.de/einrichtungen/ag/ag-netz>

Berlin, March 3rd, 2022

Thank you very much for your plan to publish a suitably revised version of our paper. We again thank the three reviewers for their work and their support of our manuscript. In the letter below, we reproduce the new comments by reviewers #2 in full and reply point by point. New changes in our manuscript are highlighted in red, except minor modifications.

Reviewer #2

A few questions related to Reviewer 1's comments: Why would the "the limiting free energy of a naked ion" be the limit for umbrella sampling? Shouldn't the limiting [i.e. 'evaporated state'] be the ion with [equilibrium] co-transferred water? Additionally, in any case why would ΔU be greater for the hydrated ion (water to vapor) than the bare ion (vapor to water) in the umbrella sampling results? The authors' insights regarding TI vs. US would be very interesting here.

In our previous reply we said that "TI data is more trustworthy, since it converges to the analytical free energy corresponding to a naked ion for $z = -3$ nm", this statement reflects the fact that the final state of the TI simulations is non-hydrated. Note that the ion in the umbrella simulations starting in the gas phase is non-hydrated as well, so we would expect that if the WHAM equilibration of the umbrella simulations worked, we should obtain a free energy in the gas phase that would agree with the analytical free energy of a naked ion. But we do not, as shown in Supplementary Figure 6. This is why we consider our umbrella simulations with the simulation times used by us as less trustworthy compared to the TI simulation results. We added some short comments on why we consider our TI simulations more trustworthy in the revised version of the manuscript.

The actual TI data significantly overshoots the 'shifted analytical curve' and the TI "fit" at $z < 2.3$ nm, as described (and well explained) by the authors. Would this TI data follow the solid black curve when $z < -3$ nm?).

This is indeed what we expect, as we now say in the revised version of the manuscript.

The approach [of performing umbrella sampling simulations by using a hydrated ion] suggested by Reviewer 1 was successful in previously published liquid/liquid simulations and showed little hysteresis. I presume this would also be the case for liquid/vapor.

Indeed, we would expect that if the simulations for each position of the umbrella potential would be equilibrated, then the free energy generated from the WHAM analysis should be reliable. But

since the equilibration of the hydration state in the vapor phase takes about 100 ns, as shown in Fig. 2g in the main text, each umbrella simulation should be longer than at least a microsecond. Again, we do not say that umbrella simulations are per se not able to obtain reliable free energy profiles for our system, we are saying that one would need much longer umbrella simulations than we used. The TI simulations are in that respect more yielding to an analysis of the effect of long equilibration times since the free energy is calculated at one fixed distance from the interface. In the revised version of our manuscript, we added a statement that the umbrella simulation trajectories are much shorter and therefore the results are not as reliable as the TI results.

Simulating a $\text{Cl}(\text{H}_2\text{O})_n^-$ complex in vacuum could be a computationally inexpensive approach toward describing the evaporated hydrated ion far from the liquid/vapor interface. A limiting case for the $z < -2.x$ region, where the liquid/ion complex interaction is very small.

Simulating a $\text{Cl}(\text{H}_2\text{O})_n^-$ complex in vacuum would quickly lead to dehydration of the ion, until a state is achieved where the water chemical potential in the vapor phase and the hydration layer are equal. So this does not help. Rather, we simulated the chloride ion at a fixed separation of $z = -7$ nm from the interface in the presence of saturated vapor, in Fig. 2g of the main text one sees that the ion becomes hydrated by about 7 water molecules after 100 ns.

The authors have satisfactorily addressed (2). That water transfers to the vapor phase one-at-a-time and transfers Cl^- with significant co-transfer of solvating water suggests interesting spaces in between, Δq and $\Delta \text{species}$.

We totally agree with the reviewer, we are currently investigating different ion species including divalent ions.

Thank you, the authors have satisfactorily addressed (3).

We are happy that the reviewer agrees with our reply.

A similar comment was given by Reviewer 3. Thank you for this addition. I agree with the authors' force field selection and interpretation. Commentary that justifies a model's selection and its limitations is important and significantly strengthens the Methods section. The authors have satisfactorily addressed (4).

We thank the reviewer for his positive comments.

The authors have satisfactorily addressed (5).

We are happy that the reviewer agrees with our reply.

This clarification is very much appreciated. The TI approach and justification for omitting data in the cubic fit is justified in the text but as mentioned, was difficult for me to parse as presented. I would suggest an attempt to clarify this aspect of the writing but leave the decision of whether to revise to the authors.

Additionally, the question I inserted into Reviewer 1's comments: Do the authors think that the TI data would follow the solid black curve when $z < -3$ nm? That is, would the TI data 'kink' at this point?

We have added some comments in the revised version of our paper. In particular, we say that we expect the TI data to follow the solid black curve for $z < -3$ nm if (and only if) the TI simulation time is less than needed in order to achieve a fully equilibrated hydration layer. We also say that

the TI data would be expected to join the solid black curve smoothly, since for a finite system there cannot be non-analyticities in the observables.

We hope that we could answer all questions by reviewer 2 and hope that our paper can be accepted for publication.

Yours sincerely,

the authors